# Disparate acidification and calcium carbonate desaturation of deep and shallow waters of the Arctic Ocean

Yiming Luo[1], Bernard P. Boudreau[1] & Alfonso Mucci[2]

The Arctic Ocean is acidifying from absorption of man-made $CO_2$. Current predictive models of that acidification focus on surface waters, and their results argue that deep waters will acidify by downward penetration from the surface. Here we show, with an alternative model, the rapid, near simultaneous, acidification of both surface and deep waters, a prediction supported by current, but limited, saturation data. Whereas Arctic surface water responds directly by atmospheric $CO_2$ uptake, deeper waters will be influenced strongly by intrusion of mid-depth, pre-acidified, Atlantic Ocean water. With unabated $CO_2$ emissions, surface waters will become undersaturated with respect to aragonite by 2105 AD and could remain so for $\sim 600$ years. In deep waters, the aragonite saturation horizon will rise, reaching the base of the surface mixed layer by 2140 AD and likely remaining there for over a millennium. The survival of aragonite-secreting organisms is consequently threatened on long timescales.

[1] Department of Oceanography, Dalhousie University, 1355 Oxford Street, Halifax, Nova Scotia, Canada NS B3H4J1. [2] Department of Earth and Planetary Sciences, McGill University, Montreal, Quebec, Canada H3A 0E8. Correspondence and requests for materials should be addressed to B.P.B. (email: bernie.boudreau@dal.ca).

The Arctic region (Fig. 1) is warming twice as fast as elsewhere in the world[1], causing rapid sea-ice cover and thickness decline[2] and, according to various emission scenarios, this may lead to an ice-free Arctic Ocean by the end of this century[3]. The anthropogenic $CO_2$ causing this warming is also acidifying the oceans[4], which will modify the carbonate chemistry of the Arctic Ocean[5] and pose a serious threat to $CaCO_3$-producing organisms[6–8]. The Arctic Ocean is already exhibiting signs of acidification[6,7]; this will be strengthened on the disappearance of the sea ice[3,5,8], which will enable more efficient ocean–atmosphere gas exchange.

Pre-industrial waters of the deep Arctic basins were largely supersaturated[9] with respect to aragonite and calcite (the two most common $CaCO_3$ polymorphs in the marine environment), due to modest metabolic $CO_2$ production in these waters from low rates of organic matter re-mineralization[10,11]; nevertheless, decreasing deep-water $CaCO_3$ saturation states and undersaturation with respect to aragonite (the more soluble of the two $CaCO_3$ polymorphs) now occur[11]—also see data below. Considering the potential consequences of these changes to this major, but vulnerable, marine eco-system, it is paramount to be able to predict accurately the future evolution of the carbonate system of the Arctic Ocean. To do so, our knowledge should include the means by which Arctic waters will acidify and the duration of those conditions. Models for the carbonate system of the Arctic Ocean, particularly its deep waters, are, however, surprisingly scarce[3,12–14]. Yamamoto et al.[3] examined surface water pH conditions to the year 2100, but they did not address acidification of the deep waters. Steinacher et al.[13] and Frölicher

and Joos[14] predict, using a three-dimensional (3D) circulation-biogeochemical model that, under two reasonable $CO_2$ emission scenarios, acidification will spread into the deeper waters from initially acidified surface waters.

Arctic acidification is, however, likely to be different than pictured by these latter models. Data presented by Miller et al.[11] and also calculated from Key et al.[15,16] (see below) indicate the recent development and expansion of aragonite undersaturation in deep water, a phenomenon not featured in the predictions made in Frölicher and Joos[14]. In addition, work by Bates et al.[17] and Cai et al.[18] questions if surface Arctic waters can act as a continuous source of $CO_2$ for acidification of the deep waters. Furthermore, the results in Steinacher et al.[13] and Frölicher and Joos[14] appear somewhat inconsistent with the known circulation in the Arctic Ocean[19–22] (Fig. 1). Specifically, Atlantic Ocean water (yellow arrows) enters the Eurasian Basin at intermediate depth, mostly between ∼300 and ∼600 m, via the Barents Sea and the East side of the Fram Strait, with half returning to the Atlantic through the West side of the Fram Strait. The rest of this input enters the Amerasian Basin, to loop back once again into Eurasian Basin, leaving by the West side of the Fram Strait and the Robeson Channel to the Northwest of Greenland. This element of the Arctic circulation indicates that pre-acidified waters from the Atlantic Ocean (Nordic Sea) should enter Arctic waters at intermediate depths, but this seems to be absent in Frölicher and Joos[14]. In addition, the halocline of the Arctic Ocean is particularly strong[19], and effective propagation of acidified surface waters into the deep will necessitate some breakdown of that stratification. Evidence for pycnocline

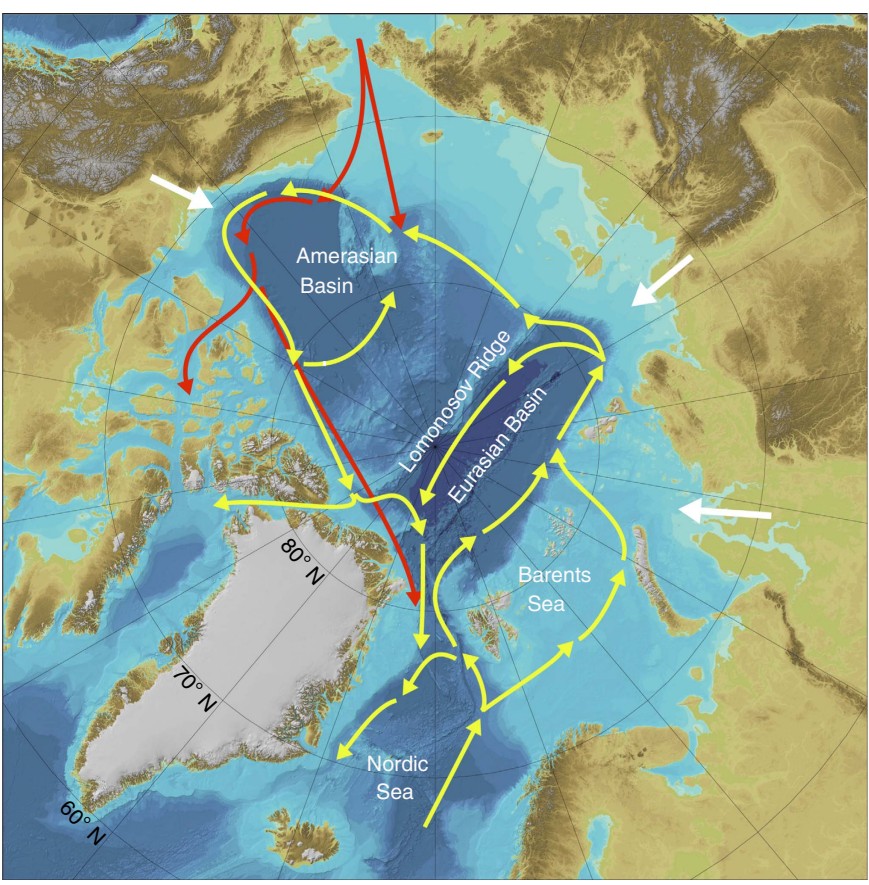

**Figure 1 | Map of the Arctic regions and major currents/water flows.** IBCAO[23] map of the two major deep basins, the Amerasian Basin and the Eurasian Basin, as separated by the Lomonosov Ridge. Red and yellow arrows indicate schematically Pacific and Atlantic inflow waters, respectively[19–22]. White arrows represent major river run-off from continents to the Arctic Ocean[19].

breakdown is currently enigmatic: sea-ice melt and increased river discharge should intensify that stratification, whereas the disappearance of sea ice may increase the fetch and wind-driven mixing, which could erode the halocline. The case for breakdown of the stratification is consequently moot.

On the basis of the observed trends in carbonate chemistry[11,15,16] and the physics of the Arctic Ocean noted in the above paragraph, we present predictions from an alternative carbonate-dynamics box model. With unabated $CO_2$ emissions, our model results forecast rapid, near simultaneous, acidification of both surface and deep waters, conditions that will persist for many hundreds to thousands of years. The surface waters will acidify directly by $CO_2$ uptake from the atmosphere, but the intrusion of mid-depth, pre-acidified, Atlantic Ocean water (or possibly from stronger thermohaline overturning) will lower aragonite saturation of the deep waters to the point of undersaturation and lead to a bottom-to-top rise of the saturation horizon. The duration of this undersaturation is a threat to Arctic calcifying organisms.

## Results

**Model implementation.** The boxes used in our model (Fig. 2) represent both the Amerasian Basin (4,250 m maximum depth)

and the Eurasian Basin (4,750 m maximum depth), separated by the Lomonosov Ridge. Each basin is divided into surface (0–200 m), intermediate (200–700 m) and deep (>700 m) water boxes, and there exist flows between these boxes and from external sources, that is, rivers, as well as the Atlantic and Pacific Oceans. Boxes are also included to account for sediment accumulation and benthic $CaCO_3$ dissolution. The justification for the use of a box model is provided in the Methods section.

Our model equations account for and predict the changes in total dissolved $CO_2$ ($\sum CO_2$) and carbonate alkalinity (CA) in each of these boxes, as forced by the increasing $CO_2$ in the atmosphere and in water feeding into the Arctic. The model subsequently calculates the pH (National Bureau of Standards) and the aragonite saturation state—see the Methods section with regard to the pH scale. Time-varying $\sum CO_2$ and CA of the Pacific surface and high-latitude Atlantic waters that enter the Arctic are obtained from the output of a previously published global carbon-system model[24,25]. The atmospheric $pCO_2$ is also calculated by that same model, in a procedure similar to that used by Spall[21].

These calculations start with initial water flows, concentrations and parameter values (Methods) believed to have been in place during pre-industrial times, that is, pre-1850 AD (all dates are AD

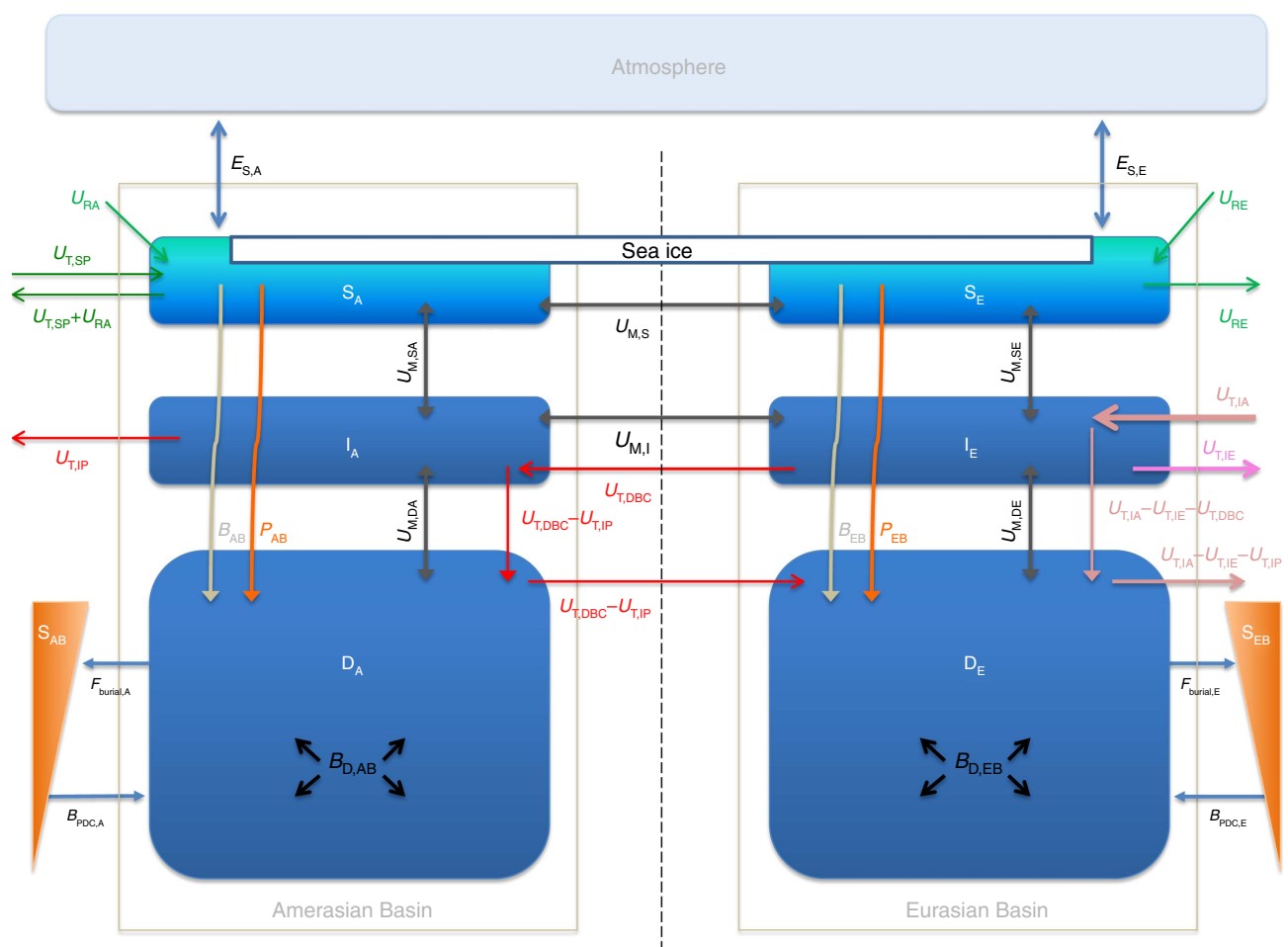

**Figure 2 | The box model for the carbonate system of the Arctic Ocean.** The model divides the Arctic Ocean into a collection of six boxes, that is, surface Amerasian Basin ($S_A$), intermediate Amerasian Basin ($I_A$), deep Amerasian Basin ($D_A$), surface Eurasian Basin ($S_E$), intermediate Eurasian Basin ($I_E$), deep Eurasian Basin ($D_E$), as well as an atmospheric box and two sediment reservoirs ($S_{AB}$ and $S_{EB}$) connected to the deep water boxes. Water flows between boxes are designated by capital $U$'s, the capital $E$'s indicate gas exchanges, the $B$'s stand for internal fluxes of $CaCO_3$ and the $F$'s are the fluxes of $CaCO_3$ to the sediments. Subscripts refer to the particular ocean (A and E), the different depth boxes (S, I and D) and the nature of the flux, for example, T for thermohaline, M for mixing, D for dissolution and so on. In addition, DBC stands for deep boundary current and PDC abbreviates previously deposited carbonate.

and that suffix is dropped hereafter), as calculated by correcting and averaging compiled $\sum CO_2$ and total alkalinity (TA) data[15,16] (Supplementary Figs 1 and 2, and Supplementary Table 1). CA is calculated from TA through standard methods (that is, from pH, total boron concentrations and appropriate dissociation constants). Flow values were obtained as explained in the Methods section, and other assigned parameters can be found in Supplementary Table 1.

The increase in $pCO_2$, which acidifies the oceans, is driven with an extended version of the IS92a emission scenario. That extended IS92a scenario was used in Boudreau et al.[24] and is described and displayed in Supplementary Fig. 3. We emphasize that this is not the original IS92a scenario, but a version modified to be consistent with emissions data and atmospheric $CO_2$ levels to 2010 and extended to predict emissions over the next millennia. This scenario may overestimate future emissions, but precaution demands that we focus on the worst possible case.

Carbon dioxide absorption from the atmosphere into the surface Arctic Ocean (dark blue $E_{S,A}$ and $E_{S,E}$ in Fig. 2) is thought to be hindered by persistent ice cover[3]. Thus, as global warming reduces the extent and duration of ice cover, $CO_2$ adsorption may increase, although that has been debated[17,18], as noted above; nevertheless, we include ice melting and corresponding increased $CO_2$ uptake in our model. The evolution of the ice cover with warming has been modelled with both the fast and slow melting scenarios by Yamamoto et al.[3] (Supplementary Fig. 4). Both scenarios were tested and our results were identical in both cases, hence, we only report the results with fast melting.

**Model output and observed data.** Whereas the model results are in the form of $\sum CO_2$ and CA, we are primarily interested in the evolving saturation state of the Arctic waters with respect to aragonite, $\Omega_a$

$$\Omega_a = \frac{[Ca^{2+}][CO_3^{2-}]}{K_{sp}^*} \qquad (1)$$

where $[Ca^{2+}]$ is the calcium concentration, $[CO_3^{2-}]$ is the carbonate ion concentration and $K_{sp}^{\star}$ is the stoichiometric solubility product under in situ conditions. A similar equation applies to calculation of the saturation state with respect to calcite. $[Ca^{2+}]$ is calculated from the known salinities of the Arctic Ocean basins. Note that, irrespective of the water chemistry, the aragonite (and calcite) saturation state changes with depth (pressure) in response to the increasing solubility of carbonate minerals[25,26], that is, $K_{sp}^{\star}$ increases with depth (pressure).

The aragonite saturation horizon ($Z_{sat}$) is the depth above which the waters are supersaturated with respect to aragonite ($\Omega_a > 1$) and below which waters are undersaturated ($\Omega_a < 1$) and in which aragonite will dissolve. To predict the position of that horizon, we coupled our box model to an explicit formula[24,27] for $Z_{sat}$,

$$Z_{sat} = Z_{sat}^1 \ln\left(\frac{[Ca^{2+}][CO_3^{2-}]}{K_{sp}^1}\right) \qquad (2)$$

where $Z_{sat}^1$ is a characteristic depth calculated from the solubility equations[25,26] for aragonite and $K_{sp}^1$ is the value of $K_{sp}^{\star}$ at 1 atm. The short derivation of this latter equation is repeated for completeness in the Methods; note that equation (2) is not dependent on the form of the model being used, that is, a box versus a 3D model.

Central to our interests are the relative roles of atmospheric $CO_2$ forcing ($E$) and the input of waters from the Atlantic and Pacific Oceans ($U_{T,IA}$, $U_{T,SP}$ in Fig. 2) in changing the pH and $\Omega_a$ of the Arctic Ocean. To facilitate this analysis, we created a hypothetical reference state wherein the $pCO_2$ in the Arctic atmosphere follows our prescribed $CO_2$ emissions scenario (Fig. 3a and Supplementary Fig. 3), but $\sum CO_2$ and CA of the inflowing Atlantic and Pacific waters do not change with time, labelled 'constant source' in our figures; these particular inflows are set to year 2010 values. In contrast, a more realistic model results by allowing Atlantic and Pacific water chemistries to change as dictated by the evolving atmospheric $CO_2$, and those model results are presented with the label 'variable source'.

The model predicted evolution of the pH (NBS) in the surface and deep waters is illustrated in Fig. 3, assuming our extended IS92a $CO_2$ emissions scenario (Fig. 3a) and constant productivity for both organic matter and $CaCO_3$ (Methods). The surface results are obtained without freshening from melting ice and increased run-off and must be accepted with this caveat.

pH is an inexact measure of the carbonate chemistry of marine waters; consequently, we have also calculated the aragonite (orange lines) and calcite (green lines) saturation states of the surface water, as shown in Fig. 3d,e. Surface waters will become undersaturated with respect to aragonite ($\Omega_a < 1$), with a minimum between 0.75 and 0.675 attained close to the year 2200 in both basins.

To test our model, we compare our output with available saturation data. Figure 4a is a contour plot of the aragonite saturation state for the Amerasian Basin calculated from the data compilations of Jutterström et al.[28] and Key et al.[15,16], as illustrated in Supplementary Figs 1 and 2. These data indicate a rise of about 500 m over the 1995–2009 period. Figure 4b reproduces the saturation isopleths provided by the 3D model used by Frölicher and Joos[14], which are essentially flat over the 1995–2009 data time series. Finally, Fig. 4c displays our model predictions of $\Omega_a$ isopleths for that same period and reveals that the isopleths are sloped by an amount similar to the data in Fig. 4a, with an upward displacement of ~300 m, based on the $\Omega_a = 1.2$ isopleth. Our model predicts slightly more acidic deep water in year 1995 than the data, but the observed and modelled isopleths below 2,000 m are roughly at the same depth in 2009.

Finally, Fig. 5 displays our prediction of the long-term evolution of the aragonite saturation horizon (red lines) and that made by Frölicher and Joos[14] (black lines). These plots are explained and analysed below.

## Discussion

The predicted pH will decline rapidly in both Arctic basins over the next 200 years (Fig. 3b,c). Our surface pH predictions (black lines in Fig. 3b,c) are consistent with the changes forecast in Yamamoto et al.[3] between the years 2000 and 2100, but are slightly higher in value, due to our slightly higher initial pH values (Methods). Amerasian surface waters will acidify a bit earlier, but the Eurasian waters will attain lower pH values, as they are initially more acidic. The surface pH minimum occurs near the year 2200 in both basins, as our emissions scenario (Fig. 3a and Supplementary Fig. 3) reaches a maximum shortly before that year. Thereafter, surface pH will slowly recover towards pre-industrial levels on a timescale >1,500 years. The surface pH values are not sensitive to the mid-depth input of acidifying Atlantic waters, that is, dashed versus solid black lines in Fig. 3b,c.

Deep-water pH in both basins is strongly dependent on the introduction of pre-acidified Atlantic waters, that is, in Fig. 3b,c, compare the dashed blue line generated with constant $\sum CO_2$ and CA in the entering Atlantic water and the solid blue line with acidifying Atlantic water. Deep-water pH would change by <0.2 units if the atmosphere was the only source of anthropogenic $CO_2$ to Arctic waters; instead, as the entering Atlantic waters acidify, they drive a pH decrease of ~0.8 unit. The minimum pH will

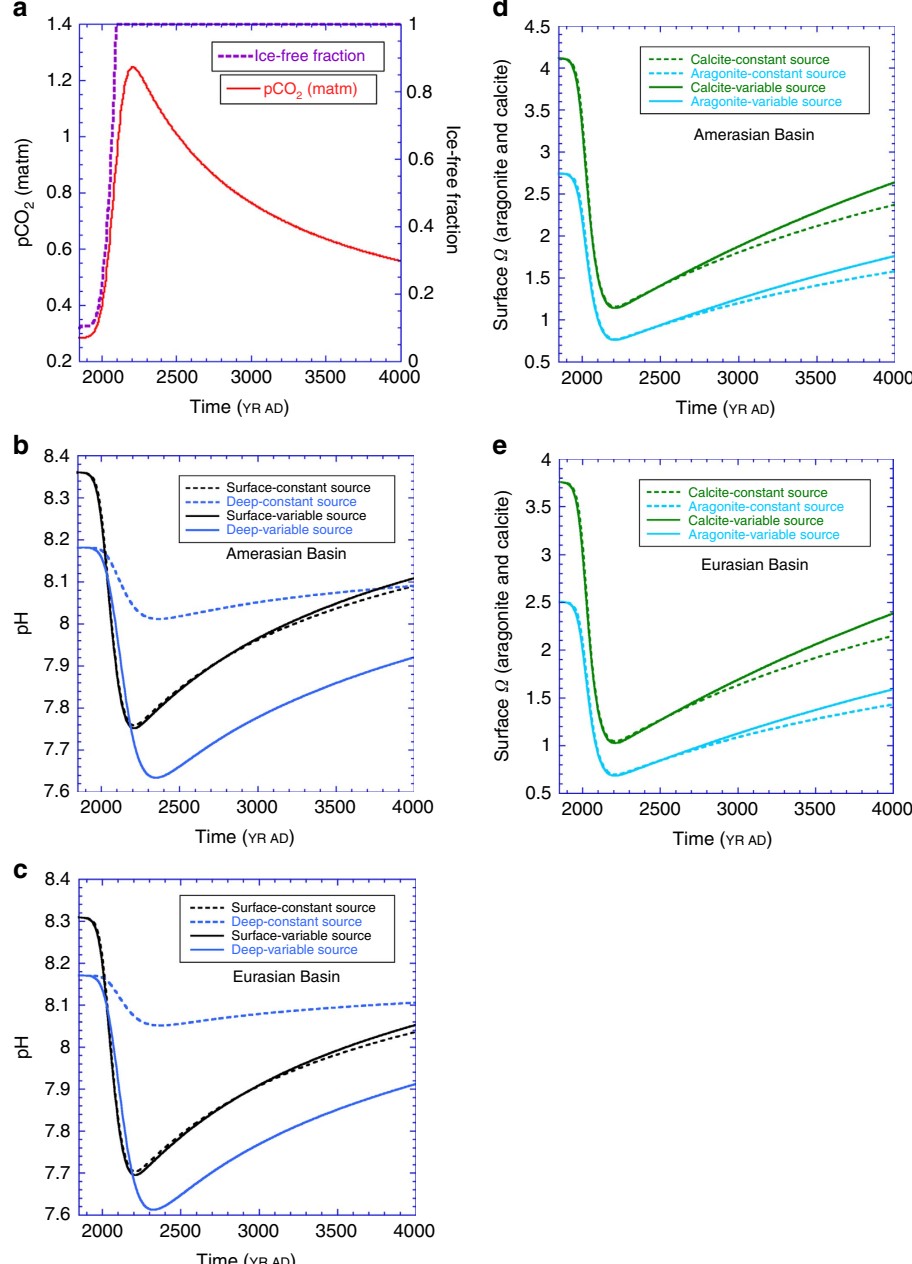

**Figure 3 | Calculated pH and surface CaCO₃ saturation of the Arctic basins.** (**a**) The pCO₂ created by the extended IS92a carbon emission scenario[24] and the ice-free fraction of the surface Arctic Ocean[3] as used in these calculated results. (**b,c**) pH in surface box and deep box (average) in Amerasian Basin and Eurasian Basin, respectively. (**d,e**) Carbonate saturation state for both aragonite and calcite in surface box in Amerasian Basin and Eurasian Basin, respectively.

occur slightly later in the deep waters than at the surface, that is, closer to the year 2350. There is clearly no evidence from these results that acidified surface waters penetrate effectively into deep waters.

With respect to the saturation state data (Fig. 4a), the $\Omega_a$ isopleths below 2,000 m rise about 500 m over the sampling period. An even greater shallowing of the $\Omega_a$ isopleths ($\sim 800$ m) is also evident in the data reported in Miller *et al.*[11] (their Fig. 10) over a similar time interval, but we chose not to use those latter data, as we could not establish the reason for discrepancies in their lower overall saturation levels compared with that calculated from the Jutterstrom *et al.*[28] and Key *et al.*[15,16] data. Nevertheless, deep acidification since year 1994 is unequivocal in both data sets.

These rises in saturation isopleths are not contained in the Frölicher and Joos[14] model results (Fig. 4b), which implies that those authors appear not have incorporated the process(es) responsible for early, deep-water, acidification/desaturation. Our box model results (Fig. 4c) provide superior data prediction of the deep-water aragonite saturation isopleths. We attribute this success to a better accounting of Atlantic deep-water penetration ($U_{T,IA}$), but it could also reflect overturning/ventilation of deep waters with surface waters, which is known to occur[29,30]. We tested this latter hypothesis and found that ventilation rates $>4.5$ Sv would be needed to quantitatively acidify the deep waters. Given that this amount currently seems excessive, we chose not to pursue this mechanism at this time;

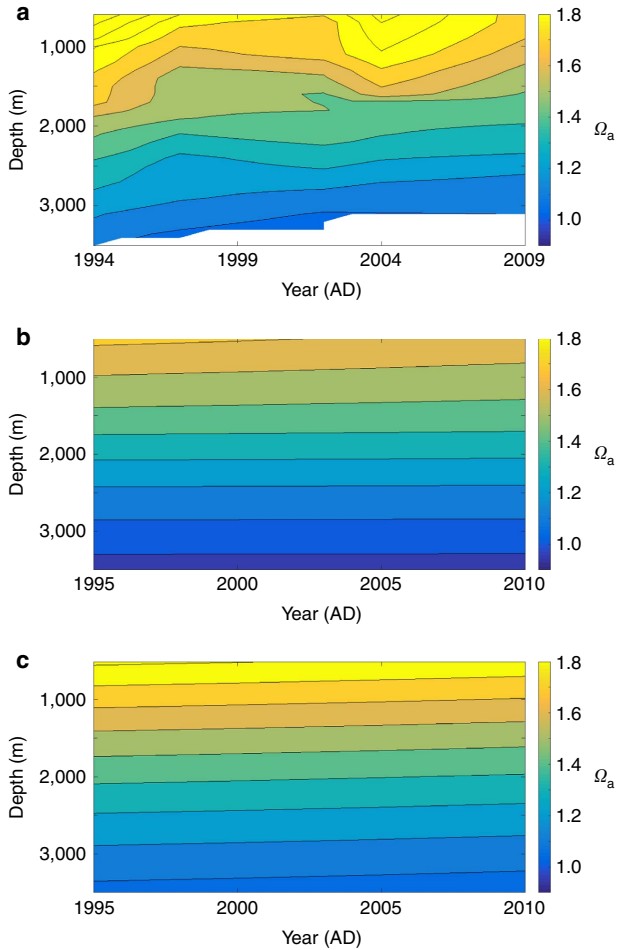

**Figure 4 | Contour plots of aragonite saturation in the Arctic Ocean.**
Lines are isopleths of constant $\Omega_a$. (**a**) Calculated from the data in Jutterstrom et al.[28] and Key et al.[15,16]. (**b**) Calculated with a 3D ocean-circulation-biogeochemistry model by Frölicher and Joos[14]. (**c**) Calculated with equation (2) and the carbonate chemistry from our model (Fig. 2). (Note the 1-year time shift in the plot for data compared with the plots of the model results; this is an artefact of the contour.m function in Matlab.)

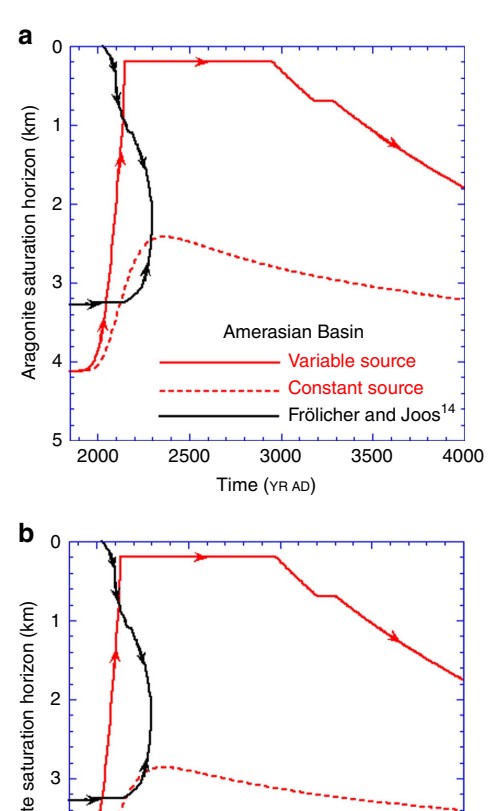

**Figure 5 | Predicted evolution of the aragonite saturation horizon.** The arrows indicate the direction of change on a given line segment. (**a**) The position of $Z_{sat}$ in Amerasian Basin with our model (red lines) and that predicted in Frölicher and Joos[14] (black line). (**b**) The position of $Z_{sat}$ in the Eurasian Basin from our model (red lines) and that predicted in Frölicher and Joos[14] (black line).

nonetheless, this possibility should be explored further through observations and modelling before it is dismissed.

Over a longer timescale, that is, millennia, the chemical evolution of the Arctic basins can be encapsulated by considering the position of the aragonite saturation horizon. Our prediction from equation (2) for the location of $Z_{sat}$ and that from Frölicher and Joos[14] are illustrated in Fig. 5a,b as red and black lines, respectively. To aid in the interpretation of this figure, arrows have been added to show the direction of time on a line segment. The black line for Frölicher and Joos[14] must be read in two parts: for the period of years 1850–2030, $Z_{sat}$ is positioned at about 3,250 m depth; it then also appears at the base of the mixed layer by the year 2030, as surface waters acidify. As time moves forward, the near-surface $Z_{sat}$ moves deeper, while the deep $Z_{sat}$ initially stays constant at 3,200 m. Around the year 2200, the deeper $Z_{sat}$ position also starts to migrate upward, indicating increasing deep-water acidification. The two horizons meet at about year 2325 at a depth of ∼2,400 m, and the entire deep Arctic Ocean is undersaturated thereafter. This complex bi-directional evolution is primarily driven by downward penetration of acidified waters from the surface.

In contrast, our model, as represented by the solid red lines in Fig. 5a,b, predicts that deep-basin waters were supersaturated to

depths closer to 4,000 m in pre-industrial times. Anthropogenically linked undersaturation then appears before year 2000 in the bottom waters of both basins because of deep Atlantic inflow, as well as in the surface waters (as seen in Fig. 4a). The saturation horizon should then move monotonically upward, without any indication of significant penetration from above the halocline. We predict that $Z_{sat}$ will reach the base of the surface mixed layer by about the year 2275 and stay there until ∼2970, which means that the intermediate and deep Arctic Ocean will become completely undersaturated with respect to aragonite slightly before the time predicted by Frölicher and Joos[14].

The red dashed lines in Fig. 5a,b illustrate the evolution of a hypothetical $Z_{sat}$ if the entering North Atlantic waters were not to acidify beyond today's conditions (constant source). A comparison of the solid and dashed red lines shows again that continued acidification and input of Atlantic waters dominates saturation changes in the deep Arctic Ocean and cannot be neglected. This conclusion is reached, however, without taking into account the effects of a possible weakening of $CO_2$ uptake by waters sinking in the Atlantic sub-polar region[31].

From an ecological point of view, what is very disturbing in Fig. 5 is that intermediate and deep waters will remain undersaturated with respect to aragonite for a period greater

than a millennium. The likelihood that the ecology of deep-water carbonate-secreting organisms could withstand that length of stress and remain unchanged is decidedly slight[32].

In conclusion, our investigation of future Arctic acidification-desaturation reveals that both the Amerasian and Eurasian Basins will experience severe undersaturation with respect to aragonite and that acidification of the surface and intermediate/deep waters will be driven by the atmosphere and intermediate-depth Atlantic Ocean water inputs, respectively. Thus, our results reiterate the pressing need to fully apprehend the role of inter-oceanic flows in changing the carbonate chemistry of the Arctic Ocean. In addition, once established, aragonite undersaturation will persist for up to 500 years in surface waters (Fig. 3) and several millennia in deep and intermediate waters (Fig. 5), conditions that constitute a threat to the survival of Arctic calcifying organisms at all depths.

## Methods

**Rationale for a box model.** Why did we use a box model rather than a 3D ocean-circulation-biogeochemical model? We thought it prudent to address this point explicitly. The model used yields results for the carbonate system that are very similar to 3D models for the Atlantic–Indian–Pacific Ocean system[21]. Why does a box model do as well as 3D models? Because the gradients, both vertical and horizontal, of $\sum CO_2$ and CA are small within the oceans, including the Arctic[15,16]. Supplementary Figs 1 and 2 provide evidence of the small gradients in the carbonate variables in the Arctic. For example, below 200 m, the $\sum CO_2$ changes presently from about 2,135 to 2,165 mM in going from the mid Eurasian Basin to the mid Amerasian Basin, a distance of about 2000 km. Thus over 2,000 km, the change is only 1.3%. As to the future, our model output indicates that the gradients will remain of this order. If the gradients are weak, the spatial resolution provided by 3D models is not necessary to answer the type of questions we present in our paper. In addition, 3D-biogeochemical ocean models are notoriously difficult to integrate very far into the future, and box models do not suffer from that problem.

**pH scale differences.** Our initial dissolved inorganic carbon and CA concentrations are slightly different than those in Yamamoto et al.[3], resulting in moderately higher surface water pH (NBS), that is, 8.38 (Amerasian Basin) and 8.28 (Eurasian Basin) for our model versus 8.22 in Yamamoto et al.[3] In comparison, Lansard et al.[33] report surface water pH (total proton scale) values anywhere between 7.95 and 8.26 for the southeastern Beaufort Sea; conversion to the NBS scale would raise these values by about 0.09 units, so that our values are within that range.

**Model water flows, parameters and initial concentrations.** Choosing appropriate parameters for our model is a major challenge, due to lack of constraints on overturning circulation, diffusion, ocean productivity, freshwater input and many other active processes operating in the Arctic. We set our pre-industrial conditions using values that are generally consistent with current knowledge (Supplementary Table 1), and we address our reasoning below.

Within our model (Fig. 2), horizontal flows/exchanges are derived from refs 19–22 and consist of 0.8 Sv (where $Sv = 10^6 \, m^3 \, s^{-1}$) of Pacific surface water entering and leaving the surface Amerasian box (olive green arrows in Fig. 2), while 6 Sv of Atlantic water (pink arrow) enter the intermediate Eurasian box, with 3 Sv (purple arrow) returning to the Atlantic from that same box. The intermediate Amerasian box is ventilated by 2 Sv of circulation from the intermediate Eurasian box (red 'in' arrow for that box), with 1 Sv ventilating and connecting the deep Amerasian box with the deep Eurasian box (red 'down' arrow) and another 1 Sv leaving the intermediate Amerasian box (red 'out' arrow) to balance the outflow through Davis Strait. The deep Eurasian box is ventilated by 1 Sv of water sinking from the intermediate Eurasian box (pink arrow) and 1 Sv returned from the deep Amerasian box (red arrow). Between surface and intermediate boxes, we allow 1 Sv of mixing, while the mixing between intermediate and deep boxes is set to 2 Sv. In addition, Amerasian and Eurasian basins are connected by mixing between surface and intermediate boxes both at 1 Sv. These added mixing flows are not taken from the literature, but are instead determined from the model's need to reproduce the pre-industrial, steady-state, carbonate chemistry.

The state-of-art estimate of the total net primary productivity (NPP) in the entire Arctic region is about 608 TgC per year in 2011, which is ~30% increase in 15 years, concordant with ~9% per decade sea-ice retreat. Assuming that ice cover is 90% of the total Arctic surface Ocean at preindustrial and the current ice cover is 70%, the ice-free open ocean area has increased 200% so far. Therefore, we assume that the total NPP in the pre-industrial Arctic was 240 TgC per year, that 6% occurred in the Arctic Basins and 10% of this basinal NPP sinks into the deep Arctic Basins. (The latter two percentages are highly ill-constrained.) The total export productivity in preindustrial open Arctic Ocean is thus no more than 120 GmolC per year. (This estimate neglects possible direct carbon transfers from

the shelves to the deep waters.) This is a small export flux, which has little effect on the carbonate system of the Arctic. With a slightly lower PIC:POC ratio of ~0.3 compared with today's average conditions in the world's ocean, and assuming that the NPP in the Amerasian and the Eurasian Basins are proportional to their surface area, we obtained the export productivity, P, values shown in Supplementary Table 1.

Note CA was obtained from TA after a correction for the borate ion contribution.

**Derivation of equation (2).** Equation (2) is not derived from a box model, but can use the carbonate ion concentration calculated from a box model to get the ACD. This is explained in refs 24,25 of the main text, but, for clarity, we repeat the logic. The thermodynamic mass-action law for $CaCO_3$ dissolution must hold at the saturation horizon $Z_{sat}$:

$$K_{sp}^c(Z_{sat}) = [Ca]_D [CO_3]_D \quad (3)$$

where $K_{sp}^c(Z_{sat})$ is the stoichiometric solubility product of $CaCO_3$ at depth $Z_{sat}$ and $[Ca]_D$ and $[CO_3]_D$ are the concentrations of calcium and carbonate ions in the deep ocean, respectively.

$K_{sp}^c(Z_{sat})$ is a function of temperature, pressure and solution chemistry (unlike true thermodynamic constants), and consequently ocean depth, which can be expressed as an implicit transcendental function of these variables. Nevertheless, Boudreau et al.[24] have found that it is possible to express the dependence of $K_{sp}^c(Z_{sat})$ on ocean depth, $Z$, as a simple exponential to a reasonable degree of accuracy, that is

$$K_{sp}^c(Z) = K_{sp}^1 \exp[\alpha Z] \quad (4)$$

where $K_{sp}^1$ is the solubility product at 1 atm pressure and at the temperature and salinity of the deep ocean, and $\alpha$ is an attenuation constant derived from empirical fits. $K_{sp}^1$ can be calculated from a polynomial fit to the experimental determinations. Values of $\alpha$ are derived by assuming that the pressure correction at each temperature is calculable from standard formulas.

Substitution of equation (3) into equation (4) at $Z_{sat}$ produces equation (2). Equation (2) is thus independent of the type of model we used in our paper. To use it, we need to specify $[CO_3]_D$, when the other parameters and variables are known. $[CO_3]_D$ can come from data or from a model, and it can be spatially and temporally variable. However, current data for the oceans, and in this case the Arctic Ocean, show that below a few hundred metres, $[CO_3]_D$ is almost a constant with depth, because $(\sum CO_2)$ and TA (and so CA) are very weak functions of depth (Supplementary Figs 1 and 2) and, to first-order approximation, $[CO_3]_D \approx CA - \sum CO_2$. As a result, we can use our box model's prediction of $[CO_3]_D$ to plug into equation (1) to obtain $Z_{sat}$.

**Code availability.** The FORTRAN computer code used to generate our results (ARCTIC.f) is available from the corresponding author. This code is not necessarily user-friendly, but its main sections and subroutines are documented.

**Data availability.** All data used in this paper (Fig. 4 and Supplementary Figs 1 and 2) were previously published and available from the cited sources. Computer-generated results can be reproduced with the code listed above, under code availability.

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

## Acknowledgements

B.P.B. and A.M. thank the Natural Sciences and Engineering Council of Canada for their financial support. Dan Kelley (Dalhousie U) provided informative consultations about water circulation in the Arctic.

## Author contributions

Y.L. and B.P.B. conceived the project; A.M. provided knowledge of Arctic flows and carbonate chemistry; Y.L. conducted the numerical calculations; and Y.L., B.P.B. and A.M. analysed the results and then wrote the paper.

## Additional information

**Competing financial interests**: The authors declare no competing financial interests.

**How to cite this article**: Luo, Y. *et al.* Disparate acidification and calcium carbonate desaturation of deep and shallow waters of the Arctic Ocean. *Nat. Commun.* 7:12821 doi: 10.1038/ncomms12821 (2016).

