## [Peer Review File · Nature Communications]

Reviewers' Comments:

Reviewer #1 (Remarks to the Author)

I find this to be a very interesting concept, illustrating that one doesn't need a complicated numerical model to assess the future evolution of ocean acidification in the Arctic Ocean. That also the intermediate and deep waters largely get under-saturated with respect to aragonite within a not too distant future is important. I think this deserves to be published but there are a few things that can be improved.

There are observations that illustrate the penetration of anthropogenic CO₂ into the intermediate and deep waters that support this modeling result. I suggest referring to at least Ericson, et al., J. Geophys. Res., 119, doi:10.1002/2013JC009514, 2014, to strengthen the output of this model. I would urge the authors to look for references of the observations used in the journal Earth Syst. Sci. Data. The way it now reads in the supplement "what could be found in the Lamont-Doherty Earth Observatory web site" does not give enough credit to the ones that have spent much time and money in collecting these data.

In the text it reads, alkalinity, total alkalinity, and carbonate alkalinity. I can't see that authors mean more than one thing with this so be consistent and use the same expression all through the ms.

On the last line of page 1 it reads that the Arctic Ocean has a weak capacity to neutralize carbonic acid because it has low alkalinity (and giving two references). The surface waters of the Arctic Ocean has relatively high alkalinity for its salinity as the river runoff entering it has a high alkalinity, a mean of around 1000 micro mole per kg!

Finally there needs to be more information behind equation 1. How one can achieve a variable saturation horizon from a box model of three depth layers needs more explanation.

Reviewer #2 (Remarks to the Author)

Luo et al. investigate future acidification of the Arctic ocean with a simple box model and present results that indicate simultaneous acidification of surface and deep waters. While the results are interesting, there are fundamental issues which, in my view, prohibit the publication of this paper in Nature Communications, at least in its current form.

There is a mismatch between the claims that the authors make and the evidence they provide. The applied model is too simple and the presented analysis of the results is too brief in order to thoroughly show something new, which previous studies might have missed. The applied model is a simple carbonate dynamics model, which does not account for the effects of reduced sea-ice cover, warming and freshening of surface waters, and potential changes in circulation or vertical stratification, which are expected to occur with amplified climate change in the Arctic. Also, the

external source of waters entering from the Atlantic, which the authors identify as a main mechanism, is obtained from a simple box model and does not account for changes in stratification, circulation, or biological activity in the Atlantic, like an Earth system model would do. This, together with a very brief and superficial description of the model and methods, as well as the lack of a comprehensive model validation, are my major concerns with this study.

Further, the authors make some fairly strong and general statements in the paper, which are not backed very well. For example, in the abstract they write that "waters will desaturate in about 100 years and remain so for ~600 years, whereas deep waters will remain undersaturated for millennia". As explained above, the applied model is rather static and probably not appropriate for such long time scales. But also, and more importantly, they only consider one (outdated) scenario of future CO₂ emissions. Since the evolution of ocean acidification strongly depends on future CO₂ emissions, a range of emission scenarios, including mitigation options, must be considered to make such a general claim.

In the following I will elaborate on those key issues and additional concerns I have with this paper.

Major concerns

First paragraph: "...that will create an ice-free Arctic by the century's end.": Again, this depends on future CO₂ emissions and is not valid as a general statement.

Second paragraph: The authors write that "Models for that carbonate system of the Arctic Ocean are, however, surprisingly limited" without providing any arguments why they would be limited. Yamamoto et al. only examined surface water conditions, which may be a limitation of the study but not necessarily of the model. Steinacher et al. and Frölicher and Joos found different results than the authors, which is not a good argument to claim limitations of their model. Further, the results of Steinacher et al. and Frölicher and Joos do not imply that there is no lateral transport of acidified water from the Atlantic Ocean that decreases the saturation state of deep waters. But they show in their simulations that the surface is undersaturated first, mainly due to freshening and increased gas exchange due to sea ice retreat, both mechanisms which are not well represented in the presented model, but backed by observational evidence (Yamamoto-Kawai et al., 2009).

The results of this study are based on the IS92a CO₂ emission scenario, which was published by the IPCC in 1992. In the meantime there have been many updates of the scenarios used for the last two IPCC reports (SRES and RCP). To investigate the impact of CO₂ emissions on future ocean acidification it is necessary to consider a range of plausible emission pathways.

Particularly with a simple model that is computationally cheap to run, I would have expected results from multiple, more recent emission scenarios.

The authors argue that the application of a box model is appropriate, because the gradients are weak. However, they don't provide much information to support this and, more importantly, that this will be the case in the future, particularly on longer timescales.

Minor issues

Abstract, line 1: I think 'absorption' is the correct term, not 'adsorption'. Also in other parts of the manuscript.

Abstract, line 2: Models don't 'argue' - this sentence seems awkward.

Results, para 4: "prescribed" instead of "proscribed".

Methods: "Beaufort" instead of "Beauford".

Model parametrization: "constraints" instead of "constrains".

Acknowledgments: "provided" instead of "provide".

Reviewers' comments: **Our replies in Red**

Reviewer #1 (Remarks to the Author):

I find this to be a very interesting concept, illustrating that one doesn't need a complicated numerical model to assess the future evolution of ocean acidification in the Arctic Ocean. That also the intermediate and deep waters largely get under-saturated with respect to aragonite within a not too distant future is important. I think this deserves to be published but there are a few things that can be improved.

There are observations that illustrate the penetration of anthropogenic CO₂ into the intermediate and deep waters that support this modeling result. I suggest referring to at least Ericson, et al., J. Geophys. Res., 119, doi:10.1002/2013JC009514, 2014, to strengthen the output of this model.

We added this. Thank you.

I would urge the authors to look for references of the observations used in the journal Earth Syst. Sci. Data. The way it now reads in the supplement "what could be found in the Lamont-Doherty Earth Observatory web site" does not give enough credit to the ones that have spent much time and money in collecting these data.

We now reference Jutterstrom et al. (2010) and Key et al. (2010; 2016)

In the text it reads, alkalinity, total alkalinity, and carbonate alkalinity. I can't see that authors mean more than one thing with this so be consistent and use the same expression all through the ms.

Actually there are 2 different quantities, total alkalinity and carbonate alkalinity. The carbonate alkalinity is calculated from the total with a set of corrections, mostly from the borate ion contribution to the latter. The two parameters are now properly and fully identified.

On the last line of page 1 it reads that the Arctic Ocean has a weak capacity to neutralize carbonic acid because it has low alkalinity (and giving two references). The surface waters of the Arctic Ocean has relatively high alkalinity for its salinity as the river runoff entering it has a high alkalinity, a mean of around 1000 micro mole per kg!

This was changed to refer to the deep waters and to indicate it is a relative statement only.

Finally there needs to be more information behind equation 1. How one can achieve a variable saturation horizon from a box model of three depth layers needs more explanation.

Equation 1 is not derived from a box model, but can use the carbonate ion concentration calculated from a box model to get the CCD. This is explained in Boudreau et al. (2010, GRL and 2010, GBC); however, for clarity, we repeat the logic. The thermodynamic mass-action law for CaCO_3 dissolution must hold at the saturation horizon z_{sat} :

$$K_{\text{sp}}^c(z_{\text{sat}}) = [\text{Ca}]_{\text{D}} [\text{CO}_3]_{\text{D}} \quad (\text{R1})$$

where $K_{\text{sp}}^c(z_{\text{sat}})$ is the conditional/stoichiometric solubility product of CaCO_3 at depth z_{sat} and $[\text{Ca}]_{\text{D}}$ and $[\text{CO}_3]_{\text{D}}$ are the concentrations of calcium and carbonate ions in the deep ocean, respectively. $K_{\text{sp}}^c(z_{\text{sat}})$ is, in general, a function of temperature, pressure, and solution chemistry (unlike true thermodynamic constants), and consequently ocean depth, which can be expressed as an implicit transcendental function of these variables, e.g., Mucci [1983] and Millero [1983]. Nevertheless, Jansen et al. [2002], Tyrrell and Zeebe [2004], and Boudreau et al. [2010, GRL] have found that it is possible to express the dependence of $K_{\text{sp}}^c(z_{\text{sat}})$ on ocean depth as a simple exponential to a reasonable degree of accuracy, i.e.,

$$K_{\text{sp}}^c(z) = K_{\text{sp}}^0 \exp(\alpha z) \quad (\text{R2})$$

where K_{sp}^0 is the stoichiometric solubility product at 1 atm pressure and at the temperature and salinity of the deep ocean, and α is an attenuation constant derived from empirical fits. K_{sp}^0 can be calculated from the formulas available in Hain et al. [2015]. Values of α are derived by assuming that the pressure correction at each temperature is calculable from the formulas (change in partial molal volumes and compressibilities for the dissolution reaction) in Millero [1995].

At the ACD and CCD, the rate of calcium carbonate dissolution at the sediment-water interface is balanced by the rate of deposition of biogenic tests. Boudreau et al. [2010 GRL] translated this definition into an equation expressing the position of the ACD or CCD as a function of its controlling variables and parameters, starting with the mathematical statement:

$$\frac{B}{A_{\text{D}}} - R_{\text{cc}} = 0 \quad (\text{R3})$$

where B is the (total) export rain of CaCO_3 into the deep oceans, A_{D} is the area of the deep seafloor, and R_{cc} is the rate of dissolution per unit area of deep seafloor. Equation (R3) constitutes a statement of mass conservation of CaCO_3 at each point on the 2D line defined by the ACD or CCD on the seafloor. Boudreau [2013, GRL] has shown that R_{cc} is (largely) controlled by the linear rate of boundary-layer mass-transfer of the carbonate ion,

$$R_{cc} = k_c (C_{sat} - [CO_3]_D) \quad (R4)$$

where k_c is the mass transfer coefficient (rate constant), and C_{sat} is the carbonate ion concentration that would be in equilibrium with $CaCO_3$ at z_{cc} . C_{sat} can be calculated as

$$C_{sat} = \frac{K_{sp}^c(z_{cc})}{[Ca]_D} \quad (R5)$$

where $K_{sp}^c(z_{cc})$ is the stoichiometric solubility product at the depth of the ACD or CCD. This equation is valid because spatial gradients in $[Ca]_D$ can be neglected, as $[Ca]_D$ is nearly conservative (proportional to salinity) in the water column.

Substitution of equations (R2), (R4) and (R5) into equation (R3) gives that

$$\frac{B}{A_D} - k_c \left(\frac{K_{sp}^0 \exp(\alpha z_{cc})}{[Ca]_D} - [CO_3]_D \right) = 0 \quad (R6)$$

or, solving for z_{cc}

$$z_{cc} = \alpha^{-1} \ln \left(\frac{B[Ca]_D}{K_{sp}^0 A_D k_c} + \frac{[Ca]_D [CO_3]_D}{K_{sp}^0} \right) \quad (R7)$$

So, this is equation (1) of the paper, and it is independent of the type of model we used in our paper. To employ it, we need to specify $[CO_3]_D$, when the other parameters and variables are known. $[CO_3]_D$ can come from data or from a model, and it can be spatially and temporally variable. However, current data for the oceans, and in this case the Arctic Ocean, show that below a few hundred meters, $[CO_3]_D$ is almost a constant with depth, because $\sum CO_2$ and total alkalinity (TA) (and carbonate alkalinity, CA) are very weak functions of depth (and $[CO_3]_D \approx CA - \sum CO_2$) – see the figures in the Supplementary Information. As a result, we can use our box model's prediction of $[CO_3]_D$ to plug into equation (R7), i.e., equation (2), to obtain z_{cc} (the CCD)!

Reviewer #2 (Remarks to the Author):

Luo et al. investigate future acidification of the Arctic ocean with a simple box model and present results that indicate simultaneous acidification of surface and deep waters. While the results are interesting, there are fundamental issues which, in my view, prohibit the publication of this paper in Nature Communications, at least in its current form.

There is a mismatch between the claims that the authors make and the evidence they provide.

See the rebuttals to the points raise in the Major Concerns. (below)

The applied model is too simple and the presented analysis of the results is too brief in order to thoroughly show something new, which previous studies might have missed.

The notion that our model is “too simple” implies that only complex models can explain the world around us, and we would disagree. Saying that our analysis is too brief is arguably a valid point, so we have included more comparisons between available data and the results from our model and the currently available 3D model (Frölicher and Joos, 2010) – see below. We think that these results conclusively show the 3D model is currently lacking something.

The applied model is a simple carbonate dynamics model, which does not account for the effects of reduced sea-ice cover, warming and freshening of surface waters, and potential changes in circulation or vertical stratification, which are expected to occur with amplified climate change in the Arctic.

Our model does indeed account for the effects of reduced sea-ice cover – look at Figures 3A and S4. We did not take into account the warming and freshening, but these are primarily surface water effects, and we focus on deep water. Even without warming and surface freshening, our results for surface are entirely similar to those in Yamamoto et al. (2012). Thus, is there some indication we are neglecting a first-order effect on the carbonate system? We think not.

Also, the external source of waters entering from the Atlantic, which the authors identify as a main mechanism, is obtained from a simple box model and does not account for changes in stratification, circulation, or biological activity in the Atlantic, like an Earth system model would do. This, together with a very brief and superficial description of the model and methods, as well as the lack of a comprehensive model validation, are my major concerns with this study.

The input model, i.e., Boudreau et al. (2010, GBC), was thoroughly reviewed for GBC, and we were made to prove that our model results were entirely compatible with those of the 3D and multi-box models listed in Archer et al. (2009, Annu. Rev. Earth Planet. Sci. 2009. 37:117–34) in terms of atmospheric CO₂ prediction, oceanic pH and CaCO₃ burial. There is no reason to believe that using this model to determine total CO₂ input from the Atlantic is fundamentally

faulty. In addition, the Boudreau et al. (2010, GBC) model explores extensively the effects of changing CaCO_3 and organic matter export (which was its primary goal), so we do know if it should be included and what the effects are. We did test the effects of changing overturning circulation in that model and found them to be second order on a time scale of interest, i.e., thousands of years; thus, they were not included in that paper.

Further, the authors make some fairly strong and general statements in the paper, which are not backed very well. For example, in the abstract they write that "waters will desaturate in about 100 years and remain so for ~600 years, whereas deep waters will remain undersaturated for millennia". As explained above, the applied model is rather static and probably not appropriate for such long time scales. But also, and more importantly, they only consider one (outdated) scenario of future CO_2 emissions. Since the evolution of ocean acidification strongly depends on future CO_2 emissions, a range of emission scenarios, including mitigation options, must be considered to make such a general claim.

Our model is not "static", but instead includes the most important time evolving processes that effect the deep waters (sea-ice changes and changing chemistry of the Atlantic). Our surface water pH and saturation results are quantitatively similar to those in Yamamoto et al. (2012) for the surface waters, where the remaining missing/static processes would be most important. Furthermore, the reviewer mis-interprets the CO_2 emissions scenario we employed, as explained below.

This reviewer paints our manuscript as a contest between a box model and a fully dynamic 3D model. Based on that premise, he/she argues the failures and short-comings of box models versus a 3D model, and that box models should not be trusted (in comparison). But all of that is a "red herring". We don't argue that box models are superior, only more convenient for our purposes in our specific case. We don't deny that box models have limitations and we don't advocate not using 3D models.

What we really argue is that the Frölicher and Joos 3D model does not currently explain the known data (**see the figures below and the revised paper**). As such, the deep-water predictions of the 3D model are entirely suspect *at this time*. If corrective action is taken with the 3D model, then we believe it can be made to correspond to the actual data (Fig. A below); at that point, its predictions of the deep will probably be far superior to the box model. Nevertheless, that is not the case right now. That is a major reason to publish our paper.

In the following I will elaborate on those key issues and additional concerns I have with this paper.

Major concerns

First paragraph: "...that will create an ice-free Arctic by the century's end.": Again, this depends on future CO2 emissions and is not valid as a general statement.

We altered the offending sentence, as it did not refer to our own work, i.e.,

"The Arctic region (Fig. 1) is warming twice as fast as elsewhere in the world¹, causing rapid sea-ice cover and thickness decline² and, under various emission scenarios, this may lead to an ice-free Arctic Ocean by the end of this century³.

We hope that is sufficient.

Second paragraph: The authors write that "Models for that carbonate system of the Arctic Ocean are, however, surprisingly limited" without providing any arguments why they would be limited. Yamamoto et al. only examined surface water conditions, which may be a limitation of the study but not necessarily of the model.

We altered the sentence to read:

"Models for the carbonate system of the Arctic Ocean, particularly its deep waters, are, however, surprisingly scarce^{3,13-15}. "

We hope that clarifies the issue.

Steinacher et al. and Frölicher and Joos found different results than the authors, which is not a good argument to claim limitations of their model. Further, the results of Steinacher et al. and Frölicher and Joos do not imply that there is no lateral transport of acidified water from the Atlantic Ocean that decreases the saturation state of deep waters.

We never claimed that the difference between our model and that of Steinacher et al. and Frölicher and Joos was "a good argument to claim limitations of their model." We are fully aware that **one model cannot be used to disprove another model**. What we said was that Frölicher and Joos' predictions for the deep Arctic Ocean were not consistent with available data, which show increasing undersaturation in the deep waters with time. To make this point, we now include three new figures, as explained below.

During the period between submission of our original paper and the Editor's decision, we were able to make concrete comparisons between measured aragonite saturation data from the Amerasian (Canada) Basin and the results of both our model and the predictions given by the

3D model in Frölicher and Joos (2010). These results are displayed, respectively, in Figs A, B, and C, below (and now in the text as Fig. 4).

Figure A is the composite of the deep water data (> 500 m) from the repository at http://cdiac.ornl.gov/oceans/LDEO_Underway_Database/. These data are for a very limited window in time, i.e., 1994 to 2009. You will notice that there is an obvious and systematic decrease in the deep-water saturation state, given by the Omega contour labels (contoured with Matlab's *contour.m* program), over that period, despite some obvious short-term transients, near the surface. For example, the Omega contour labelled 1.2 rises about 500 m in that time interval, indicating significant acidification of the deep waters. (It would have been nice to have data from deeper within this Basin, but that is what we have to work with.)

Figure B displays the predictions from our model. If you again focus on the 1.2 contour, you will notice that we predict a rise in that level of saturation by slightly over 300 m in that period, which is due to early deep acidification. In fact, our contours correspond remarkably well with those created from the data (Fig. A), if the data is time-averaged slightly. (It cannot be expected that our model can duplicate the near-surface year-to-year transients in Fig. A because (1) a box model has an inherent time-space averaging and (2) we are ignorant of the forcings that caused the short-term transients.)

Figure C contains the predictions from the 3D model (Frölicher and Joos, 2010). You will note two important facts:

- A) The contours in the predictions do not correspond to their positions in Fig. A.
- B) The contours are essentially flat, i.e., they do not rise perceptibly between 1994-2009.

Thus, not only are the Frölicher and Joos model results inconsistent with the observed saturation state in the deep Amerasian Basin, they indicate no measurable deep acidification, in contrast to the data in Fig. A.

Figure A – Amerasian Basin Omega data. Notice the sloped isopleths of Omega (aragonite) in the deep Arctic (below 2000 m) and the noisier, but acidifying surface waters.

Figure B – Box model prediction of the data in Fig. A. Notice the sloped isopleths of Omega (aragonite) in the deep Arctic (below 2000 m), and their similarity to those in Fig. A.

Fig. C – Deep-water Omega values from the 3D model in Frölicher and Joos (2010), which has essentially flat deep water (> 1500 m) isopleths.

Our box model gives a better prediction than the earlier Frölicher and Joos 3D model because it contains or better represents one or more processes, e.g., Atlantic deep-water penetration, not present or adequately represented in the 3D model. At this point, we are giving a superior prediction of what is happening in the Arctic over the stated time period of the data and that is important for climate and ocean acidification researchers and environmental managers to know. But we are not getting these better and different results because we have a superior model, but because we consider a process - Atlantic deep-water penetration - not included or adequately represented in the earlier model.

But they show in their simulations that the surface is undersaturated first, mainly due to freshening and increased gas exchange due to sea ice retreat, both mechanisms which are not well represented in the presented model, but backed by observational evidence (Yamamoto-Kawai et al., 2009).

Our model, which does explain the deep data, shows simultaneous acidification of the deep and surface, and our model **includes** ice retreat as a forcing factor and this is stated explicitly in the manuscript. We don't have freshening because we don't model density, but instead assign it; so, this is a shortcoming of our box model. We can only offer that freshening is difficult to model, even with a fully dynamic 3D model, and that this freshening will apply to the surface acidification, which is not the primary message of our paper.

The results of this study are based on the IS92a CO₂ emission scenario, which was published by the IPCC in 1992. In the meantime there have been many updates of the scenarios used for the last two IPCC reports (SRES and RCP).

It's probably our bad writing, but we stated that we used the “**extended IS92a**” emissions scenario to drive the model. It is not the original and dated IS92a scenario. The extended IS92a scenario is described in detail in Boudreau et al. (2010, *Global Biogeochem. Cycle*, v. 24, GB4010, doi:10.1029/2009GB003654) and illustrated in Fig. D below.

Fig. D. *Extended IS92a CO₂ emissions scenario as given in Boudreau et al. (2010).*

Panel A of Fig. D gives the known emissions from 1800 to 2009 (black dots) and our fit and extension of those emissions into the future (red line), which we label the “extended IS92a” scenario. The emissions curve describes the input of CO₂ to the combined atmosphere-ocean system, hence discounting land uptake; this explains why the model forcing in Figure D – panel A is slightly lower than the actual emissions data based on estimated fossil fuel emissions. The emission scenario follows the IPCC IS92a projection for the 21st century (Figure D – Panel B), but is entirely consistent with the observed atmospheric CO₂ data from Law Dome and Mauna Loa to year 2010. The extended IS92a then uses a forward Gaussian evolution, which peaks near the year 2250. The total emission is 4025 Gt C over a period of 600 years, which leads to maximum atmospheric CO₂ concentrations of around 1400 ppm (Figure D – Panel B). This type of extension is needed because of the long period of integration of our model, i.e., thousands of years. This scenario is as accurate as anything in the IPCC reports (SRES and RCP) for the period with data. We will add this figure to the Supplementary Information for completeness.

To investigate the impact of CO₂ emissions on future ocean acidification it is necessary to consider a range of plausible emission pathways. Particularly with a simple model that is

computationally cheap to run, I would have expected results from multiple, more recent emission scenarios

Here is where we differ strongly with the reviewer. None of the other emissions scenarios has any basis beyond the year 2100. How will they be extended? Using other scenarios becomes simply a mathematical exercise. The real point of our paper is, if we allow emission to continue unabated, how bad will the situation get and we only need one scenario to show that.

The authors argue that the application of a box model is appropriate, because the gradients are weak. However, they don't provide much information to support this and, more importantly, that this will be the case in the future, particularly on longer timescales.

The figures in the Supplementary Information provide ample proof of the small gradients in the carbonate variables in the Arctic. For example, below 200 m, the ΣCO_2 changes presently from about 2135 mM to 2165 mM in going from the mid Eurasian Basin to the mid Amerasian Basin, a distance of about 2000 km. Thus over 2000 km, the change is 1.3%, and by any definition that is weak. As to the future, our model output indicates that the gradients will remain of this order.

We do wish to point out to the Editor that box models are often preferred for long time-scale integrations because they do not suffer from the instabilities inherent to the circulation portion of 3D models. Again we refrain from stating this in the paper because we do not wish to have our paper considered as a critique of 3D models, which it is not.

Minor issues

Abstract, line 1: I think 'absorption' is the correct term, not 'adsorption'. Also in other parts of the manuscript.

Agreed and modified as requested.

Abstract, line 2: Models don't 'argue' - this sentence seems awkward.

Changed to "Model results argue ..."

Results, para 4: "prescribed" instead of "proscribed".

Agreed and modified as requested.

Methods: "Beaufort" instead of "Beauford".

OK.

Model parametrization: "constraints" instead of "constrains".

OK.

Acknowledgments: "provided" instead of "provide".

OK.

Reviewers' Comments:

Reviewer #1 (Remarks to the Author)

The authors have improved the manuscript by their revisions that address both mine and the other reviewer's comments. There still is one thing that was missed. In the text, line 6-7 of the result section, it is stated that the carbonate alkalinity is illustrated in S2. However, it is the total alkalinity that is illustrated (or at least noted). This needs to be corrected by either changing the figure, or explaining that the carbonate alkalinity was computed from the DIC and TA of S1 and S2.

With this correction I find this manuscript suitable for publication.

Reviewer #2 (Remarks to the Author)

I thank the authors for their reply, which clarified some of my concerns. While I still disagree in some points that are being discussed, I think that in the revised manuscript they present their work in an objective way, such that the readers can judge for themselves. Therefore I can support the publication of the revised manuscript in Nature Communications.

REVIEWERS' COMMENTS:

We reply in dark red.

Reviewer #1 (Remarks to the Author):

The authors have improved the manuscript by their revisions that address both mine and the other reviewer's comments. There still is one thing that was missed. In the text, line 6-7 of the result section, it is stated that the carbonate alkalinity is illustrated in S2. However, it is the total alkalinity that is illustrated (or at least noted). This needs to be corrected by either changing the figure, or explaining that the carbonate alkalinity was computed from the DIC and TA of S1 and S2.

With this correction I find this manuscript suitable for publication.

We changed the problematic paragraph to read:

“These calculations start with initial concentrations believed to have been in place during pre-industrial times, i.e., pre-1850 AD (all dates are AD and that suffix is dropped hereafter), as calculated by correcting and averaging compiled ΣCO_2 and total alkalinity (TA) data^{16,17} – see Supplementary Figs 1 and 2, and Supplementary Table 1. CA is calculated from TA through standard methods. Flows values were obtained as explained in the Methods and other assigned parameters can be found in Supplementary Table 1.”

We hope that this resolves the matter.

Reviewer #2 (Remarks to the Author):

I thank the authors for their reply, which clarified some of my concerns. While I still disagree in some points that are being discussed, I think that in the revised manuscript they present their work in an objective way, such that the readers can judge for themselves. Therefore I can support the publication of the revised manuscript in Nature Communications.

OK